# Mitigating Intrinsic Named Entity-Related Hallucinations of Abstractive Text Summarization

**Jianbin Shen**[*2]       **Jie Liang**[†2]       **Junyu Xuan**[†1]

[1]Australian Artificial Intelligence Institute (AAII), University of Technology Sydney, Australia
[2]Visualisation Institute, University of Technology Sydney, Australia
[*]jianbin.shen@student.uts.edu.au,  [†]{jie.liang, junyu.xuan}@uts.edu.au

## Abstract

Abstractive text summarization (ATS) is both important and challenging. Recent studies have shown that ATS still faces various forms of hallucination. Our study also indicates that a significant portion of hallucinations is named entity-related. They might appear in different forms, such as mistaken entities and erroneous entity references. The underlying causes implicit in data are complex: data samples pose varying learning conditions. Despite recent research efforts dedicated to named entity-related hallucinations, the solutions have not adequately addressed the varying learning conditions posed by data. This paper aims to bridge the gap in pursuit of reducing intrinsic named entity-related hallucinations. To do so, we propose an adaptive margin ranking loss to facilitate two entity-alignment learning methods to tackle them. Our experiment results show that our methods improve the adopted baseline model on automatic evaluation scores. The human evaluation also indicates that our methods jointly reduce the intrinsic named entity-related hallucinations considerably compared to the adopted baseline model.

## 1  Introduction

Abstractive text summarization (ATS) has improved considerably in recent years, attributed to the advancement of encoder-decoder (a.k.a. seq2seq) modeling (e.g., Vaswani et al., 2017). The language modeling using innovative pre-training methods (e.g., Lewis et al., 2020; Zhang et al., 2020) has further led to the significant performance gain on n-gram overlap-based metrics (e.g. ROUGES). But recent studies (e.g., Ji et al., 2022; Maynez et al., 2020) have found generative summaries prone to various forms of factual problems known as hallucinations.

Despite the research efforts leading to hallucination reduction and factuality improvement (e.g., Lukasik et al., 2020; Sridhar and Visser, 2022;

Zhao et al., 2022), some hallucinations remain a challenge to ATS. Named entity-related hallucinations (NERHs) are among them. Our study has also found that summarization of long text sequences (e.g., CNN DailyMail or CNNDM in short) incurs high occurrences of NERHs compared to other types of factual problems (e.g., negation related). NERHs can be divided into intrinsic and extrinsic. The former includes the cases where a named entity hallucinated in a summary is mentioned in the source document, while the latter occurs when a summary includes a novel named entity not in the source document. We are interested in the intrinsic NERHs as the main source of NERHs. Among the intrinsic NERHs, entity-entity hallucinations are often observed. For example, given the source document from the CNNDM test set below,

> *"... Since civil war ..., 310,000 people have been killed, the Syrian Observatory for Human Rights said Thursday. ... estimate by the U.N. of at least 220,000 dead. ..."*.

A pre-trained model fine-tuned with the CNNDM training data may generate the following summary,

> *"U.N.: More than 310,000 people have been killed in Syria ..."*.

The summary mistakes the Syrian Observatory for Human Rights for U.N. We believe that the cause is rooted in the model's misaligning of entities with their contexts in terms of sentences.

The other form of intrinsic NERHs is entity-reference hallucinations. For example, given the following source document,

> *"... with an eight-month-old baby, ... Savannah Guthrie ... help her precious baby girl Vale drift off. When ... mother-of-one discussed an Australian father's tip for getting his baby to sleep ..."*.

The fine-tuned model generates the summary,

> *"The ... mother-of-one discussed ... tips for getting his eight-month-old daughter to sleep ..."*.

As seen, "her" is mistaken for "his" in the summary. We think that the cause could lie in the reference similarity in the latent space of the model.

Researchers have proposed entity-aware approaches to tackle intrinsic NERHs (e.g., Xiao and Carenini, 2022; Bi et al., 2021). Meanwhile, the illustrated examples suggest that the models could fail to learn from data to distinguish positive candidates (e.g., her) from negative ones (e.g., his) in the latent space. Margin ranking loss is well-known for tackling such problems. For example, Chen et al. (2021) incorporate a margin ranking loss to address extrinsic entity hallucinations. The classical margin ranking loss treats all samples identically, but various forms of intrinsic NERHs are implicit in data with complicated linguistic structures and might interplay at both lexical and contextual levels. Data samples thus pose varying learning conditions (easy or difficult) for a model. And yet it is difficult to predefine and categorize them in modeling. It is desirable for methods adaptive to the variant learning conditions at a fine granularity. Perhaps missing such adaptive capacity will have limited the efficacy of the prior works.

To bridge the gap in tackling our hypothesized causes of the illustrated examples, we develop an adaptive margin ranking loss to facilitate two learning methods to jointly mitigate the intrinsic NERHs.

In summary, our contributions are:

1. We propose an adaptive margin ranking loss function. The function incorporates a named entity span-based distance-regularized intersection over union (ESpan-DIoU) metric to derive adaptive scaling variables of a chosen base margin. The ESpan-DIoU provides a well-defined scalar space permissible for our learning purpose.

2. By utilizing the adaptive margin ranking loss, we further propose an entity-sentence alignment method to tackle entity-entity hallucinations and an entity-reference alignment method to mitigate entity-reference hallucinations. The methods automatically fit data of

variant learning conditions to tackle the hallucinations of interest respectively and holistically.

3. We integrate our methods with a BART encoder-decoder. We also fine-tune the BART alone as a baseline for comparison analysis. We experiment with our methods using CN-NDM and XSum datasets. The experiment results show that our methods improve the adopted baseline model on automatic evaluation scores. Further human evaluation also indicates a noticeable reduction of the intrinsic NERHs.

## 2 Related Work

Factual issues or hallucinations are challenging to ATS despite recent progress. Many methods have been proposed in recent years to tackle the challenges. Typically, contrastive learning methods are adopted to address the factual issues caused by exposure bias or sample imbalance (e.g., Cao and Wang, 2021; Wu et al., 2022). The recent progress in summarization evaluation research also inspires new learning objectives (e.g., Wan and Bansal, 2022; Pernes et al., 2022) and question-answer-based approaches (e.g., Gunasekara et al., 2021; Nan et al., 2021b). Additionally, hallucinations in generated summaries prompt post-editing correction methods such as Cao et al. (2020) and Balachandran et al. (2022).

Meanwhile, realizing entity-related hallucinations as one of the main factual issues, researchers have also developed entity-aware methods. One idea is to encode entity-aware representations for expressiveness (e.g., Bi et al., 2021). To improve named entity matches between the generated summaries and the source documents, Xiao and Carenini (2022) propose a named entity span copy from source documents based on generative likelihood estimation with a global relevance classification task. Nan et al. (2021a) introduce a summary-worthy entity classification on the entities occurring in both the source documents and reference summaries. To address named entity-relation hallucinations, Lyu et al. (2022) introduce entity-relation generators with entity-consistency and relation-consistency learning objectives. Observing the source-summary entity aggregation phenomenon in which the named entities are replaced by more general scoped descriptions, González

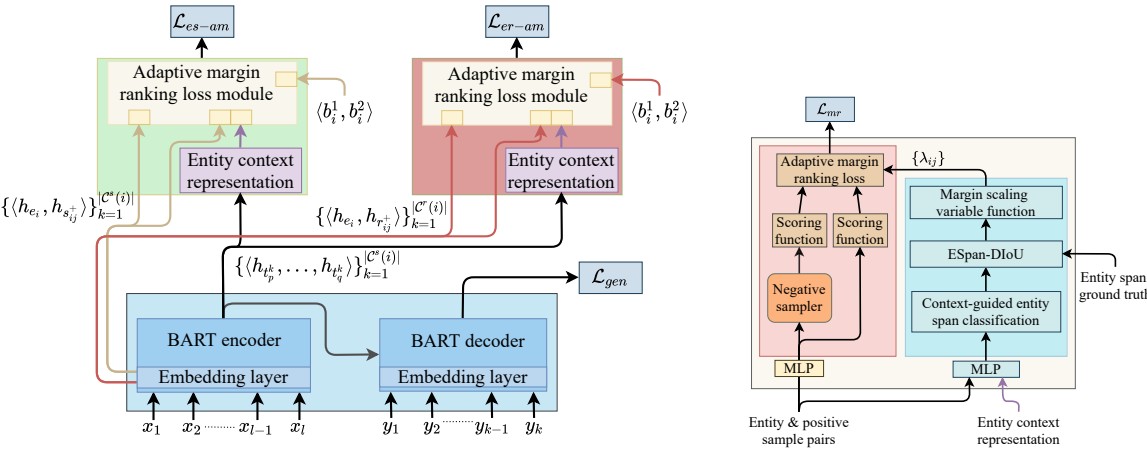

(a) Extended BART encoder-decoder.

(b) Adaptive margin ranking loss module.

Figure 1: (a) Architecture of a BART encoder-decoder (blue-ish block) extended with an entity-sentence alignment method (green-ish block) and an entity-reference alignment method (carmine block). The two alignment methods are internally based on an adaptive margin ranking loss module (milky block) and an entity context representation module (purple-ish block). (b) Adaptive margin ranking loss module consists of two key submodules, the margin ranking loss with adaptive capacity (pinkish block) and the margin scaling variable submodule (sky-blue block). MLPs are used to reduce dimensionality.

et al. (2022) introduce aggregation-awareness to ensure the aggregations are factually aligned with the aggregated entities. Introducing guiding signals, Zhang et al. (2022) propose an entity coverage control method by prepending an entity coverage precision value to the encoder's inputs while Narayan et al. (2021) chain named entities from the reference summary and prepend them to the decoding inputs. Post-editing approaches (e.g., Chen et al., 2021; Lee et al., 2022) are also developed to reduce entity hallucination as part of strategies for improving broader factual consistency.

Methods discussed so far have not adequately considered adaptiveness to the variant learning conditions at a fine granularity. In developing our adaptive methods, we desire a real-valued margin-scaling variable function to provide the resulting margins with three properties. That is, they are proportional to the (easy or difficult) learning conditions implicit in samples; they are real values within a well-defined scalar space permissible for our learning purpose; and the base margin is preserved at the low bound of the scalar space.

## 3 Our Methods

Our methods[1] are illustrated in Fig. 1. The architecture as shown in Fig. 1a, consists of the backbone BART encoder-decoder (Lewis et al., 2020), the

---

[1]As our methods focus solely on named entities, we simply mean "named entity" whenever we use "entity" from now on.

entity-sentence alignment method (E-Sent AM), and the entity-reference alignment method (E-Ref AM). Both alignment methods utilize adaptive margin ranking loss that has the same modular structure as shown in Fig. 1b except that the E-Sent AM uses a similarity-scoring function while the E-Ref AM adopts an antisymmetric-scoring function. Two methods also share an entity context representation module. We first discuss the generic aspects of the architecture before proceeding to the specifics of our two alignment methods.

### 3.1 Encoder-Decoder Generative Model

An encoder takes an input sequence of $l$ tokens $\langle x_1, x_2, ..., x_l \rangle$ and outputs a latent sequence $Z = \langle z_1, z_2, ..., z_l \rangle$. A decoder then auto-regresses its inputs $\{y_i\}_{i=1}^m$ of $m$ tokens with the $Z$ latent states to generate the $Z$-attentive outputs $\langle \tilde{y}_1, \tilde{y}_2, ..., \tilde{y}_m \rangle$. The learning objective is maximum likelihood estimation (MLE) measured by cross-entropy loss:

$$\mathcal{L}_{gen} = -\frac{1}{m} \sum_{i=1}^m y_i \log(\tilde{y}_i), \qquad (1)$$

where $\tilde{y}_i$ and $y_i$ are the $i^{th}$ word prediction and the corresponding ground truth respectively.

### 3.2 Adaptive Margin Ranking Loss

Margin ranking (or triplet ranking) loss, first proposed by Schroff et al. (2015), can be defined as:

$$\mathcal{L}_{mr} = \max(0, sc_k^+ - sc_k^- + m), \qquad (2)$$

where $sc_k^+$ and $sc_k^-$ are positive and negative sample scores concerning an anchor sample of the $k^{th}$ sample triplet (i.e. anchor, positive and negative samples), and $m$ is a margin hyperparameter. The function aims to distance negative samples from positive ones concerning an anchor. But the scalar margin assumes that samples pose a uniform learning condition. The assumption is hardly suitable for many complex problems including ours. This gives rise to our adaptive margin ranking loss defined as:

$$\mathcal{L}_{mr} = \max(0, sc_k^+ - sc_k^- + \lambda_k \cdot m), \qquad (3)$$

where $\lambda_k$ is a scaling variable for the $k^{th}$ triplet. We now detail its applications for our two alignment methods[2].

### 3.3 Entity-Sentence Alignment Method

An entity has its contextual sentences where the entity occurs and/or is referenced in a document. We express the contextual sentences of an entity as an entity cluster as follows:

$$\langle e_i, \{s_{ij}\}_{k=1}^{|\mathcal{C}^s(i)|} \rangle, \ \mathcal{C}^s(i) \in \mathcal{C}^s \text{ and } j \in |S|, \quad (4)$$

where $e_i$ is the entity $i$, $|\mathcal{C}^s(i)|$ is the number of contextual sentences in the entity cluster $\mathcal{C}^s(i)$, $\mathcal{C}^s$ is entity cluster set, $\{s_{ij}\}_{k=1}^{|\mathcal{C}^s(i)|}$ is the ordered set of the contextual sentences, $j$ is the sentence number in the document, and $|S|$ is the total number of sentences in the document. The contextual sentences may not be adjacent. Each sentence $s_{ij}$ is further expressed by its bounding (i.e. beginning and ending) token index pair in the document as:

$$s_{ij} \doteq \langle t_p, t_q \rangle. \qquad (5)$$

We construct the entity-sentence positive samples by transforming the entity and sentences into one-to-one pairs as follows:

$$\langle e_i, \{s_{ij}\}_{k=1}^{|\mathcal{C}^s(i)|} \rangle \Rightarrow \{\langle e_i, s_{ij}^+ \rangle\}_{k=1}^{|\mathcal{C}^s(i)|}. \qquad (6)$$

The superscription $+$ denotes positive samples.

**Entity Context Representation** Different entities can have their contexts sharing a subset of sentences. This presents ambiguous learning conditions to ATS models. Instead of mutually excluding such sharing to train our model, we take an entity context-guided learning approach discussed later to take into account the impact. We first learn the

---

2[Appendix A](#) summarizes key notations used in the methods.

---

context representation. Using the contextual sentences in Eq. 4 and each sentence's bounding token index pair in Eq. 5, we can gather the token latent states of the contextual sentences from the encoder:

$$\mathbf{h}_{e_i,\langle . \rangle}^c \doteq \{\langle h_{t_p^k}, ..., h_{t_q^k} \rangle\}_{k=1}^{|\mathcal{C}^s(i)|}, \qquad (7)$$

where $\langle h_{t_p^k}, ..., h_{t_q^k} \rangle$ is the token representation sequence of the $k^{th}$ contextual sentence. To learn the context representation of the sequence, we implement a multi-filter multi-kernel multi-scale convolutional neural network (M3-CNN) described in Appendix B. Using the model, we get the resulting representation as follows:

$$h_{e_i}^c = \text{M3-CNN}(\mathbf{h}_{e_i,\langle . \rangle}^c). \qquad (8)$$

**Negative Sampler** We construct negative sample pairs by randomly drawing the samples from the positive samples of the other entities:

$$\{\langle e_i, s_{i'j'}^- \rangle\}_{k=1}^{|\mathcal{C}^s(i)|}, \\ \mathcal{C}^s(i') \in \mathcal{C}^s, \ i' \neq i \text{ and } j' \in |S|. \qquad (9)$$

We get the sample triplets from Eq. 6 and Eq. 9:

$$\{\langle e_i, s_{ij}^+, s_{i'j'}^- \rangle\}_{k=1}^{|\mathcal{C}^s(i)|}. \qquad (10)$$

**Sample Representations** For an entity, we may simply use its key token latent state from the encoder as its representation. For sentence samples, we use an M3-CNN to learn their representations. The resulting representations of the triplets can then be expressed as:

$$\{\langle h_{e_i}, h_{s_{ij}^+}, h_{s_{i'j'}^-} \rangle\}_{k=1}^{|\mathcal{C}^s(i)|}, \qquad (11)$$

where $h_{e_i}$, $h_{s_{ij}^+}$ and $h_{s_{i'j'}^-}$ are the entity, positive and negative sentence representations respectively. With sample data ready, we can now follow Fig. 1b to instantiate adaptive margin ranking loss. We start with a measure adaptive to learning conditions.

**Context-Guided Entity Span Classification** Kryscinski et al. (2020), in their FactCCX method, develop a claim (text) span classification to identify supports and mistakes predicted by their evaluation model. Such text span classification may be regarded as a test method for learning conditions. That is, easy learning conditions are implied if a model converges the classification well. It is difficult to learn otherwise. We extend the idea by integrating context guidance. To do so, we concatenate the entity, positive sentence representations

from Eq. 11 and the context representation (Eq. 8), followed by a clamped linear transformation to get the entity span prediction logits as follows:

$$\text{logits}_{e_{ij}} = \min(\max(0, W[h_{e_i}, h_{s_{ij}^+}, h_{e_i}^c]), l_{max}),$$
(12)

where $W$ is the learning parameters, $[,]$ is the concatenation operator, $l_{max}$ is the maximum document length that the encoder-decoder permits, and $\text{logits}_{e_{ij}} \in \mathbb{R}^{2 \times 1}$ is the predictions of bounding token index pair of the entity.

**Entity Span Distance-IoU** We now need a metric on the entity span classification to give rise to a scaling variable function that satisfies the aforementioned adaptive margin properties. Intersection over union (IoU) is a well-known metric for object detection in computer vision. It measures the overlapping ratio between a predicted object and the ground truth in terms of bounding boxes. It is scale-invariant and bound to $[0.0, 1.0]$. Entity span in 1D shares a similar overlapping characteristic to the bounding box in 2D. We adopt a Distance-IoU by Zheng et al. (2020) for fast learning convergence, and derive an entity span-based Distance-IoU (ESpan-DIoU) as follows.

Let the predicted $\text{logits}_{e_{ij}}$ for the entity $i$ be split into index pair $(\hat{b}_i^1, \hat{b}_i^2)$. With the ground truth $(b_i^1, b_i^2)$, an entity span IoU is defined as:

$$
\begin{aligned}
f_\cap &= \min(\hat{b}_i^2, b_i^2) - \max(\hat{b}_i^1, b_i^1), \\
f_\cup &= (\hat{b}_i^2 - \hat{b}_i^1) + (b_i^2 - b_i^1) - f_\cap + \epsilon, \\
\text{IoU}_i &= \frac{f_\cap}{f_\cup},
\end{aligned}
$$
(13)

where $f_\cap$ and $f_\cup$ are the intersection and union of the two text spans respectively, and $\epsilon$ is a small residue to avert divide-by-zero. A DIoU is then derived as:

$$
\begin{aligned}
f_\omega &= \max(\hat{b}_i^2, b_i^2) - \min(\hat{b}_i^1, b_i^1), \\
f_\rho &= ((b_i^1 + b_i^2) - (\hat{b}_i^1 + \hat{b}_i^2))^2 / 4, \\
\text{DIoU}_i &= \text{IoU}_i - f_\rho / (f_\omega^2 + \epsilon),
\end{aligned}
$$
(14)

where $f_\omega$ is the smallest width enclosing the two spans, $f_\rho$ is the squared distance of the two span centers, and $f_\omega^2$ is the squared $f_\omega$.

**Margin Scaling Variable Function** We first convert the ESpan-DIoU to a loss bound by $[0.0, 2.0]$:

$$\mathcal{L}_{\text{DIoU}_i} = 1.0 - \text{DIoU}_i.$$
(15)

To preserve a base margin at the low bound, we transform the loss to be bound by $[1.0, 3.0]$. So, the scaling variable function for the sample $j$ of the entity $i$ is defined as:

$$\lambda_{ij} = 1.0 + \mathcal{L}_{\text{DIoU}_i}.$$
(16)

**Scoring Function** The aim is to measure the relevance of the sentences to the entity. We consider this as a similarity measure using the cosine function. A positive sample score is thus defined as:

$$sc_{ij}^+ = \frac{h_{e_i} \cdot h_{s_{ij}^+}}{|h_{e_i}| \times |h_{s_{ij}^+}|}.$$
(17)

With the same formula, a negative sample score $sc_{ij'}^-$ can be acquired with the negative sample representations $h_{s_{i'j'}^-}$ instead.

**Entity-Sentence Alignment Learning Objective** With the obtained $\{\langle sc_{ij}^+, sc_{ij'}^-, \lambda_{ij} \rangle\}_{k=1}^{|\mathcal{C}^s(i)|}$, the learning objective is computed as:

$$
\begin{aligned}
\mathcal{L}_{es\text{-}am} = & \\
\frac{1}{|\mathcal{C}^s(i)|} & \sum_{k=1}^{|\mathcal{C}^s(i)|} \max(0, sc_{ij}^+ - sc_{ij'}^- + \lambda_{ij} \cdot m).
\end{aligned}
$$
(18)

### 3.4 Entity-Reference Alignment Method

This alignment method follows the same process as section 3.3 but with different data and scoring function. We focus our discussion on the difference.

Positive samples here consist of the annotated coreferences for each entity, that is, an entity referent and its references. We also express the entity references as an entity cluster as follows:

$$\{\langle e_i, r_{ij}^+ \rangle\}_{k=1}^{|\mathcal{C}^r(i)|}, \; j \in |\mathcal{C}^r(i)| \text{ and } \mathcal{C}^r(i) \in \mathcal{C}^r,$$
(19)

where $e_i$ is an entity $i$, $r_{ij}^+$ is a reference $j$ (e.g., a pronoun), $|\mathcal{C}^r(i)|$ is the number of references in the entity cluster $\mathcal{C}^r(i)$, and $\mathcal{C}^r$ is entity cluster set.

**Negative Sampler** We construct negative sample pairs by randomly drawing the samples from the positive reference samples of the other entities:

$$
\begin{aligned}
& \{\langle e_i, r_{i'j'}^- \rangle\}_{k=1}^{|\mathcal{C}^r(i)|}, \\
& j' \in |\mathcal{C}^r(i')|, \; \mathcal{C}^r(i') \in \mathcal{C}^r \text{ and } i' \neq i.
\end{aligned}
$$
(20)

The sample triplets are then given:

$$\{\langle e_i, r_{ij}^+, r_{i'j'}^- \rangle\}_{k=1}^{|\mathcal{C}^r(i)|}.$$
(21)

**Sample Representations** As with the approach taken in section 3.3, we may use the key token latent states from the encoder for entity referent and reference representations respectively. So, the representations of the triples are expressed as:

$$\{\langle h_{e_i}, h_{r_{ij}^+}, h_{r_{i'j'}^-} \rangle\}_{k=1}^{|\mathcal{C}^r(i)|}. \tag{22}$$

**Context-Guided Entity Span Classification** The classification can share the same context representation guide as in section 3.3 but concatenates it with the entity and positive reference representation pairs from Eq. 22 as follows:

$$\text{logits}_{e_{ij}} = \min(\max(0, W'[h_{e_i}, h_{r_{ij}^+}, h_{e_i}^c]), l_{max}), \tag{23}$$

where $W'$ is the learning parameters.

**Scoring Function** An entity referent and its references are functionally different from a linguistic perspective. So, in general, they are not semantically symmetric. We expect a scoring function to be able to score the antisymmetric relations encoded in their representations. To realize this, we design the function in complex embedding space as in Trouillon et al. (2016) but with a more simple formula as follows:

$$sc = \text{Re}(E_s \overline{E_o}^{\text{T}}), \tag{24}$$

where $E_s$ is the referent representation matrix, and $\overline{E_o}$ is the complex conjugate of the reference representations $E_o$ in matrix form. As the dot product of complex vector matrices may not always result in real values, the score $sc$ is taken by the real part (expressed by $\text{Re}$ operator) for the real value-based margin ranking loss. One can easily prove that the function possesses the antisymmetric property.

**Entity-Reference Alignment Learning Objective** With the obtained $\{\langle sc_{ij}^+, sc_{i'j'}^-, \lambda_{ij} \rangle\}_{k=1}^{|\mathcal{C}^r(i)|}$, the learning objective is computed as follows:

$$\mathcal{L}_{er\text{-}am} = \\ \frac{1}{|\mathcal{C}^r(i)|} \sum_{k=1}^{|\mathcal{C}^r(i)|} \max(0, sc_{ij}^+ - sc_{i'j'}^- + \lambda_{ij} \cdot m). \tag{25}$$

### 3.5 Total Learning Objective

The total learning objective consists of the three learning objectives Eq. 1, Eq. 18 and Eq. 25 as:

$$\mathcal{L} = \mathcal{L}_{gen} + \mathcal{L}_{es\text{-}am} + \mathcal{L}_{er\text{-}am}. \tag{26}$$

| Model | CNNDM Test Set[1] | | |
|---|---|---|---|
| | R-1 | R-2 | R-L |
| QA-Span/BertSumExtAbs (Dong et al., 2020) | 41.75 | 19.27 | 38.81 |
| ERPGN/BART-Base (Lyu et al., 2022) | 42.28 | 19.64 | **38.93** |
| FactPEGASUS(Zero-shot)/BART-base (Wan and Bansal, 2022)[2] | 40.98 | 18.97 | 28.90 |
| BART-base | 42.81 | 19.52 | 29.36 |
| BART-base + E-Ref AM[3] | **43.10** | **19.82** | 29.62 |
| BART-base + E-Sent AM[4] | 42.88 | 19.56 | 29.35 |
| BART-base + Dual AMs[5] | 42.81 | 19.50 | 29.36 |

Table 1: ROUGE Evaluation (with CNNDM). 1. The number of test samples from our annotation preprocessing is 11483 (out of 11490 samples). 2. Wan and Bansal (2022) does not have results on CNNDM. We use their published source code (https://github.com/meetdavidwan/factpegasus) to train and test a model using their settings except that the maximum source length and target length are changed to 1024 and 142 respectively. 3. Entity-reference alignment method. 4. Entity-sentence alignment method. 5. Dual AMs consists of both entity-reference and entity-sentence alignment methods.

## 4 Experiment and Analysis

### 4.1 Dataset

We experiment with CNN DailyMail (CNNDM) (See et al., 2017; Hermann et al., 2015) and XSum (Narayan et al., 2018). Both datasets consist of news article and reference summary pairs. XSum's summaries are more abstractive than CNNDM's in that XSum has one-sentence alike summaries while CNNDM tends to have multi-sentence summaries. We use Stanford CoreNLP software[3] to prepare our datasets (Appendix D.1).

### 4.2 Implementation

Our methods adopt a pre-trained BART-base encoder-decoder[4]. It has 6 layers, 12 attention heads, a filter size of 3072, and hidden state dimensions of 768. The key implementation is detailed in Appendix D.2. Our source code is also accessible[5].

### 4.3 Results and Analysis

We first evaluate generated summaries using ROUGE metrics (Lin, 2004), followed by recent-developed factuality consistency evaluation metrics, SummaC (Laban et al., 2022). We then con-

---

[3]https://stanfordnlp.github.io/CoreNLP.
[4]https://huggingface.co/facebook/bart-base.
[5]https://cloudstor.aarnet.edu.au/plus/s/0UhKQEoHTULankr.

| Model | XSum Test Set[1] | | |
|---|---|---|---|
| | R-1 | R-2 | R-L |
| QA-Span/BertSumExtAbs (Dong et al., 2020) | 36.86 | 14.82 | 29.70 |
| ERPGN/BART-Base (Lyu et al., 2022) | 39.60 | 16.90 | 31.74 |
| FactPEGASUS(Zero-shot)/BART-base (Wan and Bansal, 2022) | 32.97 | 11.42 | 25.41 |
| BART-base | 41.80 | 18.99 | 33.90 |
| BART-base + E-Ref AM[2] | 41.74 | 18.84 | 33.68 |
| BART-base + E-Sent AM[3] | **41.99** | **19.06** | **34.00** |
| BART-base + Dual AMs[4] | 41.87 | 18.91 | 33.77 |

Table 2: ROUGE Evaluation (with XSum). 1. The number of test samples from our annotation preprocessing is 11328 (out of 11334 samples). 2. Entity-reference alignment method. 3. Entity-sentence alignment method. 4. Dual AMs consists of both entity-reference and entity-sentence alignment methods.

| Model | $SummaC_{ZS}$ | | $SummaC_{Conv}$ | |
|---|---|---|---|---|
| | $\mu(\%)$ | $\sigma$ | $\mu(\%)$ | $\sigma$ |
| BART-base | 68.3 | 0.255 | 62.5 | 0.230 |
| E-Ref AM | 71.6 | 0.222 | 65.0 | 0.206 |
| E-Sent AM | 68.4 | 0.239 | 64.5 | 0.201 |
| Dual AMs | **71.8** | 0.231 | **66.8** | 0.200 |
| Reference | 48.5 | 0.243 | 45.6 | 0.188 |

Table 3: SummaC score distribution statistics over the 100 randomly sampled generated summaries from the CNNDM test set.

| Model | $SummaC_{ZS}$ | | $SummaC_{Conv}$ | |
|---|---|---|---|---|
| | $\mu(\%)$ | $\sigma$ | $\mu(\%)$ | $\sigma$ |
| BART-base | **10.2** | 0.190 | 23.8 | 0.036 |
| E-Ref AM | 8.9 | 0.154 | 23.5 | 0.026 |
| E-Sent AM | 7.0 | 0.123 | 23.5 | 0.029 |
| Dual AMs | 9.9 | 0.185 | **23.9** | 0.035 |
| Reference | 6.4 | 0.108 | 23.3 | 0.028 |

Table 4: SummaC score distribution statistics over the 100 randomly sampled generated summaries from the XSum test set.

duct our human evaluation of named entity-related hallucinations (NERHs) and commonly observed syntactic agreement errors. We also extend our evaluations as detailed in Appendix E.1.

**ROUGE Evaluation** Table 1 shows the ROUGE scores evaluated with the CNNDM test set. Separated by double lines, the top section of the table lists several recent ATS factuality research using the same or similar-sized backbone models, followed by our experiment section containing the high ROUGE scores. Our experiments have evaluated the fully configured dual alignment methods and two ablations – the entity-reference alignment method and the entity-sentence alignment method. We have also fine-tuned the baseline BART-base for comparison. Similarly, Table 2 shows the ROUGE scores evaluated with the XSum test set.

Compared to prior work, the models trained on our annotated datasets have outperformed mostly R-1 and R-2 metrics on CNNDM while bettering across all metrics on XSum.

Among our experiments, we see that the E-Ref AM produces better scores with CNNDM while the E-Sent AM has the edge with XSum. Would the results also suggest that the alignment methods individually outperform the combined Dual AMs in reducing intrinsic NERHs and improving the overall factuality? Recent ATS factuality evaluation studies (e.g., Kryscinski et al., 2020; Maynez et al., 2020) have found that n-gram overlap-based metrics (e.g., ROUGEs) are not sufficient for factuality assessment. To answer the above question, we first conduct an automatic evaluation of factuality

consistency as follows.

**Automatic Evaluation of Factuality Consistency** We use the SummaC for the evaluation. It has two metric flavors, $SummaC_{ZS}$ and $SummaC_{Conv}$. $SummaC_{ZS}$ is highly sensitive to extrema as pointed out by the authors. $SummaC_{Conv}$ mitigates the sensitivity by transforming entailment probabilities into a histogram to learn a score function.

We randomly sample 100 generated summaries, then use the metrics to score each summary concerning its source document, followed by computing their score distributions (i.e. mean and standard deviation statistics) as shown in Table 3 and Table 4 for both CNNDM and XSum respectively[6].

As seen, the Dual AMs method scores higher mean values than the alignment ablations with both CNNDM and XSum even though the Dual AMs may have had lower ROUGE scores respectively.

Compared to the baseline BART-base, our methods achieve better scores with CNNDM. But, with XSum, the baseline has an advantage on $SummaC_{ZS}$ scores while the Dual AMs edges ahead on the $SummaC_{Conv}$. Along with the standard deviations, the results for XSum suggest that the baseline produces some summaries with higher probabilities while the Dual AMs generates more summaries in high entailment probability bins.

We also score the reference summaries for analysis. We see that the reference summaries have the lowest scores for both datasets. This could indicate

---

[6]Appendix E.2 provides statistical significance assessment.

| Error Type | Model | | | |
|---|---|---|---|---|
| | Our Baseline | Dual AMs | E-Ref AM | E-Sent AM |
| Entity intrinsic | 17 | 7 | 10 | 11 |
| Entity extrinsic | 8 | 7 | 9 | 6 |
| *Subtotal* | *25* | *14* | *19* | *17* |
| Modifier | 0 | 0 | 2 | 0 |
| Event | 3 | 2 | 4 | 2 |
| Event-time | 5 | 2 | 5 | 1 |
| Location | 0 | 1 | 0 | 0 |
| Negation | 0 | 2 | 0 | 0 |
| Number | 1 | 5 | 3 | 3 |
| Misspelling | 2 | 5 | 3 | 1 |
| *Subtotal* | *11* | *17* | *17* | *7* |
| *Total* | *36* | *31* | *36* | *24* |

Table 5: Human evaluation of factuality on the 100 randomly sampled generated summaries (CNNDM).

| Error Type | Model | | | |
|---|---|---|---|---|
| | Our Baseline | Dual AMs | E-Ref AM | E-Sent AM |
| Entity intrinsic | 23 | 12 | 16 | 14 |
| Entity extrinsic | 1 | 1 | 2 | 0 |
| *Subtotal* | *24* | *13* | *18* | *14* |
| Modifier | 3 | 3 | 12 | 15 |
| Event | 19 | 9 | 8 | 7 |
| Event-time | 9 | 8 | 6 | 8 |
| Location | 10 | 10 | 11 | 13 |
| Negation | 2 | 0 | 1 | 1 |
| Number | 13 | 12 | 18 | 13 |
| Misspelling | 0 | 0 | 0 | 0 |
| *Subtotal* | *56* | *42* | *56* | *57* |
| *Total* | *80* | *55* | *74* | *71* |

Table 6: Human evaluation of factuality on the 100 randomly sampled generated summaries (XSum).

that the reference summaries contain significant extrinsic knowledge. The entailment model trained mainly on intrinsic data might not fit them well.

It is noticeable that the scores for CNNDM are much higher than those for XSum. As studied by Lu et al. (2020), XSum reference summaries have 35.76% and 83.45% novel unigrams and bigrams respectively compared to CNNDM's 17.00% and 53.91%. The semantic disparity of XSum between source documents and reference summaries might expose the data imbalance in training SummaC for evaluating summaries that tend to be extrinsic. This rationale agrees with the reasoning behind the low reference summary scores.

**Human Evaluation** We further assess various aspects of factuality issues that may give rise to automatic metric scores. We evaluate how well our methods reduce the NERHs of interest. We also assess a range of commonly observed syntactic agreement errors that are often the causes of

hallucinations, covering event, event time, location, number, modifier, and negation. Misspelling errors are also included. Appendix F details the defining rules for counting the hallucinations and errors.

The same 100 sampled summaries evaluated by SummaC are used. The erroneous occurrences are shown in Table 5 and Table 6 for both CNNDM and XSum respectively. Separated by double lines, each table contains the statistics of the NERHs, the syntactic agreement issues and misspellings, followed by the sums of all counts.

As shown, our alignment methods consistently reduce the intrinsic NERHs compared to the baseline for both CNNDM and XSum: the Dual AMs method reduces them considerably. Noticed that the models trained with XSum result in much fewer extrinsic entity hallucinations than those trained with CNNDM. We think that the one-sentence conciseness of XSum summarization might have limited exposure to the extrinsic entity hallucinations.

Meanwhile, Table 5 shows that the Dual AMs and E-Ref AM methods introduce more syntactic agreement and misspelling errors than the baseline while the E-Sent AM method can reduce them. The Dual AMs method has more errors on location, negation, numbers, and misspellings while the baseline results in more errors on event and event time. In Table 6, the Dual AMs method results in the least syntactic agreement errors while the baseline has much higher error counts on events. This error-type divergence might occur because the models attend to the respective features more often.

Additionally, the models trained with XSum incur much more syntactic agreement errors than those trained with CNNDM. It is worth noting that the sampled summaries from all models have few other factual errors outside our categories. So, we believe that the abstractive and extrinsic nature of XSum summaries could have contributed to more errors in events, event time, locations, and numbers.

## 5 Conclusion

This paper proposes an adaptive margin ranking loss function. We utilize the loss function to facilitate two entity alignment methods to mitigate intrinsic named entity-related hallucinations of ATS. Our experiment results and analysis show that our methods improve the adopted baseline model on automatic evaluation scores and reduce the intrinsic named entity-related hallucinations.

## Limitations

Our research in this paper is confined to news article documents. Given the data feature used, our methods are limited to exploring the mitigation of intrinsic named entity-related hallucinations. We have properly chosen the syntactic categories for our human evaluation, and have assessed the generated summaries with due diligence and as objective as possible, but assessment inaccuracy can still occur, considered as low-bound by Bayes error. We hope that the combination of ROUGE scores, SummaC metrics, and human evaluation in the paper has strengthened our overall analysis.

## Ethics Statement

To the best of our knowledge, we have attributed our work to the previous research and implementations on which this paper depends, either in the main script or in the appendices. Any changes to third-party implementations follow their license permits. Our work and experiment results are all genuine.

## Acknowledgements

This work was supported by the Australian Research Council through Discovery Early Career Researcher Award DE200100245 and Linkage Project LP210301046.

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

| Notation | Description |
|---|---|
| $\|\cdot\|$ | The total number of items or samples. |
| $\{\cdot\}$ | An ordered set of items or samples. |
| $\{\cdot\}_{k=1}^{\|\cdot\|}$ | An ordered set indexed by an indexing variable $k$ up to the total number $\|\cdot\|$. |
| $\langle\cdot,\cdot\rangle/\langle\cdot,\cdot,\cdot\rangle$ | A pair/triplet notation. |
| $\langle\cdot,...,\cdot\rangle$ | A sequence notation. |
| $[\cdot,\cdot]$ | A dimensional concatenation operator of latent representations. |
| $\langle t_p, t_q\rangle$ | A sentence bounding (i.e. beginning and ending) token indices concerning a document. |
| $e_i$ | An entity $i$. |
| $s_{ij}$ (or $s_{ij}^+$) | A sentence $j$ in a document where an entity $i$ appears and/or is referenced (as a positive sentence sample). |
| $s_{ij'}^-$ | A negative sentence sample related to an entity $i$, that is, a sentence $j'$ in a document where an entity $i$ does not appear or is referenced. |
| $r_{ij}$ (or $r_{ij}^+$) | A reference $j$ in the reference cluster of an entity $i$ (as a positive reference sample). |
| $r_{ij'}^-$ | A negative reference sample linked to an entity $i$, that is, a reference $j'$ refers to a different entity from the entity $i$. |
| $\lambda_{ij}$ | A margin scaling variable for the sample $j$ of the entity $i$. |
| $\mathcal{C}^s(i)$ | A cluster of sentences where an entity $i$ appears and/or is referenced. |
| $\mathcal{C}^s$ | A set of all entity-sentence clusters. |
| $\mathcal{C}^r(i)$ | A cluster of references to an entity $i$. |
| $\mathcal{C}^r$ | A set of all entity-reference clusters. |
| $\|\mathcal{S}\|$ | The total number of sentences in a document. |
| $h_\cdot$ | A (latent) representation of an item or sample. |
| $h_{e_i}^c$ | An entity $i$'s (latent) context representation. |
| $sc_{i\cdot}^+/sc_{i\cdot}^-$ | Margin ranking scoring function of positive/negative sample with respect to an anchor entity $i$. |
| $\mathcal{L}_\cdot$ | A learning objective or loss notation. |

Table 7: Key notation summary. We use $\cdot$ in place of item (or sample) identities for simplicity without loss of generality.

## A   Key Notation Summary

Table 7 summarizes the key used notations for their mathematical meanings and functions in our methods.

## B   M3 Convolutional Neural Network

Convolutional neural networks with multi-filters and multi-kernels are computationally more economic than Transformer-based models for rela-

tively short sequences while still capturing salient features well. They have been used to encode a representation of short text sequences (e.g. Kim, 2014; Chen and Bansal, 2018) with a single-scale max-over-time pooling. We adopt a convolutional neural network (CNN) for encoding a representation of multi-sentential entity context. We think the choice of CNN is sensible because the inputs to the CNN are the latent states from the BART encoder-decoder. The lengths of our multi-sentential contexts are somewhat in-between full-length documents and the short-text sequences seen in the aforementioned prior work. So, we extend single-scale pooling to multi-scale pooling by borrowing the idea from deep learning in computer vision. Given a padded text sequence $x$ to the pre-defined max length and a down-sampling scale factor, we can compute the number of down-sampling scales. We can also compute input and output size along the sequence length dimension at each scale. So, each convolutional block consists of a 1D convolution, a ReLU, and an adaptive 1D max-pool. A fully-connected linear layer is applied to the output of the last down-sampling scale.

## C   Super Token Representation Learning

The BART adopts a word-segmentation-based token encoding method (Sennrich et al., 2016) to deal with oversized vocabulary issues. But the NLP parsing tools produce coreference resolution on words[7]. Researchers have applied the word-level structures either to the leading tokens of words (e.g., Heinzerling and Strube, 2019) or to the aggregated token representations (e.g., Ek and Bernardy, 2020). We take the latter approach and apply a GNN-based super token representation learning, detailed as follows.

Given the tokens of a word, we construct a token graph $\mathcal{G} = (\mathcal{V}, \mathcal{E}, \mathcal{D})$. $\mathcal{V} = \{v_0, ..., v_m\}$ is the token node set. $\mathcal{E} = \{(v_0, v_j)|j \in [0, ..., m]; \langle v_0, ..., v_m\rangle\}$ is the edge set between the leading token $v_0$ and each following tokens. $\mathcal{D} = \{0, ..., m\}$ is the distance set of the subsequent tokens to the leading token. Note that we add a self-loop edge to the leading token to deal with a single-token word case. Similar to BART's positional embeddings, we build the distance embeddings of vocabulary size $|\mathcal{D}|$ but use them as

---

[7]NLP parsing tools (e.g., Stanford CoreNLP) may use word segmentation methods (e.g., Penn Treebank tokenization) to produce words for annotation.

| Dataset | Size |
|---|---|
| Train | 149634 |
| Validation | 7782 |
| Test | 11483 |

Table 8: Preprocessed CNNDM dataset sizes.

| Dataset | Size[1] | Size[2] |
|---|---|---|
| Train | 186873 | 65698 |
| Validation[3] | 10391 | 10391 |
| Test[3] | 11328 | 11328 |

Table 9: Preprocessed XSum dataset sizes. 1. Without CR annotations. 2. With CR annotations. 3. We do not annotate CRs for validation and test datasets for inference. So, they remain the same for the two-stage fine-tunings.

edge features.

We adopt a generic GNN model proposed by You et al. (2020) for learning the representations. Given the designed graph, a single-layer GNN is sufficient to learn super token representations. In a simple form, the message passing is defined as:

$$h_u'^{\mathcal{V}} = (W_g^{\mathcal{V}} h_u^{\mathcal{V}} + b_g^{\mathcal{V}}) + (W_g^{\mathcal{D}} h_u^{\mathcal{D}} + b_g^{\mathcal{D}}), \\ u \in \mathcal{N}(v_0) \quad (27)$$

where $\{W_g^{\mathcal{V}}, b_g^{\mathcal{V}}\}$ and $\{W_g^{\mathcal{D}}, b_g^{\mathcal{D}}\}$ are the linear transformation parameters of the token embedding $h_u^{\mathcal{V}}$ and the distance embedding $h_u^{\mathcal{D}}$ respectively. The message aggregation is followed as:

$$h_{\{u\}}^{\mathcal{V}} = \sum_{u \in \mathcal{N}(v_0)} h_u'^{\mathcal{V}}. \quad (28)$$

The update is then computed as:

$$h_{v_0}'^{\mathcal{V}} = h_{v_0}^{\mathcal{V}} + h_{\{u\}}^{\mathcal{V}}. \quad (29)$$

## D  Experiment Setup

### D.1  Data Preparation

We use Stanford CoreNLP to annotate coreference resolution (CR) for the training dataset. Stanford CoreNLP uses Penn Treebank tokenization (a word segmentation method) to produce words for annotation. So, the annotated training dataset has a slightly different data distribution from the original data. To maintain validation and test with the same data distribution as the annotated training data, we also use the same process to annotate the validation and test datasets without CR.

**Annotated CNNDM Datasets**  We download CNNDM datasets (train, validation, and test) using Hugging Face's datasets Python package. We use the Stanford CoreNLP parsing software to acquire coreference resolution (CR) annotations. We develop a tool to transform the CR annotations into our training dataset format of entity referents, references, and their attributes. The attributes include animacy (e.g., ANIMATE and INANIMATE), gender (e.g., MALE and FEMALE), number (e.g., SINGULAR and PLURAL), and type (e.g., PROPER

and PRONOMINAL). We build the vocabularies of these CR attributes respectively. It is worth noting that CR entities and references are also annotated with their positions concerning their sentences and the associated sentence number. So, our process computes each sentence's length and the total number of sentences. These are word-level annotations.

We preprocess the encodings[8] of the built documents using the model tokenizer[9]. The number of encoded output tokens may exceed the length limit of the model. Truncation of the exceeded tokens would break the application of the CR annotations. We therefore exclude these samples. While building the token document, we have also created a word-token map (graph). The word-token map enables runtime mapping between word indices and token indices. The preprocessed encodings also save the model's token encoding time during training runs. Table 8 lists the final preprocessed dataset sizes.

**Annotated XSum Datasets**  Similar to CNNDM preprocessing, we download XSum datasets using Hugging Face's datasets package and annotate them using the Stanford CoreNLP parsing software. We then preprocess the annotated data to create our final train, validation, and test datasets. Noticed that a small set of reference summaries in XSum has annotated CRs. It is insufficient to fine-tune the pre-trained BART-base model directly with the CR-annotated dataset. So, we create two training sets for two-stage fine-tunings, one without CR annotations and the other with CR annotations. Note that we only conduct test evaluations on the fine-tuned model in the second stage after the model is fine-tuned with our proposed methods with the CR-derived data. We list the dataset sizes in Table 9.

---

[8] The tokenization uses the byte-pair encoding method.
[9] https://huggingface.co/facebook/bart-base.

### D.2 Key Implementation

#### D.2.1 Super Token Representation Learning

The GNN model discussed in Appendix C is implemented by Pytorch Geometric package[10]. We adopt the model implementation from the package and configure it with our settings.

#### D.2.2 Adaptive Margin Scaling Variables and Base Margin

To determine a base margin, we are interested in the dynamics of the margin scaling variables. So, we add trace logic in the source code to log the maximum and minimum values of the scaling variables per epoch during training. From our trial runs, we find that the value range starts from about $[2.25, 2.99]$ and converges towards the range about $[1.99, 2.24]$ when runs are finished on an early stop criterion. Based on the observation, we set our base margin at 25.

#### D.2.3 Alignment Methods

Referents and references may consist of multiple words. To simplify the learning methods without loss of representational discriminative power, we take a super node representation learning approach similar to Appendix C. In short, we center on the keyword of an entity or its reference, and apply a GNN to aggregate its neighboring (up to) n-gram words at each side of the keyword if applicable[11]. We set n-gram to 2 in this paper based on our observation of the annotated CR data.

Our alignment methods refactor several functions from the knowledge graph learning package PyKeen[12]. The changes are related to the complex vector based model for the entity-reference alignment method, including both interaction and margin ranking loss functions. We also add a similarity interaction interface for the entity-sentence alignment method.

#### D.2.4 Entity Span Classification in Entity-Sentence AM

The classification in Eq. 12 includes sentence feature $h_{s_{ij}}$. Given the entity context vector $h_{e_i}^c$ encapsulates the possible salient feature of the sentence, we thus simplify the equation by omitting the sentence feature in the concatenated feature vector as $[h_{e_i}, h_{e_i}^c]$.

#### D.2.5 Boost Coreference Expressiveness by Annotated CR Attributes

As discussed in Appendix D.1, the CR annotation expresses the relations between an entity referent and its reference by several coreference attributes. We implement them as attribute embeddings and combine them with the coreference representations to boost representation expressiveness.

#### D.2.6 Index Mapping from Word-Level Structure to Token

The model-encoded tokens and the word-level CRs are not aligned by their sequential position indices. That is, their corresponding indices are not the same. To apply the CR-derived data to the super tokens, we have developed source codes for index mapping by utilizing the word-token map created in Appendix D.1.

#### D.2.7 Early Stop Training Criterion

We use an early-stop training approach up to the configured maximum epoch. The criterion is the ROUGE metric-based evaluation of the validation dataset. The same ROUGE metrics for test time inference evaluation are used. The training stops when the ROUGE scores remain the same for a predefined number of consecutive times. We set the early stop criterion to 4 and the max epoch is 50.

#### D.2.8 Training on Multi-GPUs

We develop a multi-GPU running procedure based on the reference runtime script[13]. We train models on two-GPU parallelism. Our GPU cards are NVIDIA A100/80GB each[14]. A configuration of Dual AMs has a model size of 581.460MB. A fine-tuning session of the Dual AMs with CNNDM on an early-stop setting takes about 94 hours. For XSum, we first fine-tune the model without CR-derived data (i.e., without using our alignment methods). It takes roughly 26 hours on an early-stop setting. We then further fine-tune the model using the alignment methods with the CR-derived data on the same early-stop setting for about 3 hours.

---

[10]https://pytorch-geometric.readthedocs.io.

[11]Imposing n-gram constraint is because entity annotations can occasionally include the restrictive clause (e.g., which clause) as a whole.

[12]https://github.com/pykeen/pykeen/tree/master/src/pykeen.

[13]https://github.com/huggingface/transformers/blob/master/examples/pytorch/summarization/run_summarization_no_trainer.py.

[14]As the used GPUs are shared resources, we are constrained to fully utilize them for larger pre-trained models.

| Setting | CNNDM | XSum |
|---|---|---|
| Maximum Length | 142 | 62 |
| Minimum Length | 56 | 11 |
| Beam No. | 4 | 6 |
| Length Penalty | 2.0 | 1.0 |

Table 10: Inference settings.

### D.2.9 Training Setting Summary

We use AdamW optimizer. The learning rate is $5e^{-5}$ with a linear decay. The weight decay is $1e^{-6}$.

### D.2.10 Inference Setting

We extract the inference settings from the pre-trained BART-base configuration as shown in Table 10.

### D.2.11 Automatic Evaluation Metrics

**ROUGE Metrics** The ROUGE automatic metrics (R-1, R-2, and R-L) are commonly used for ATS. We use ROUGE metrics implemented by Lin (2004) via a wrapper API from Hugging Face's datasets Python package. The implementation produces low, medium, and high scores for each metric. We take the high scores in our experiment evaluations.

**SummaC Metrics** We adopt SummaC source code[15] to produce our formatted outputs.

## E Extended Factuality Assessment

### E.1 Comparison to FactPEGASUS

We extend our factuality assessment to compare with FactPEGASUS/Zero-shot by both automatic metric evaluations and manual assessment[16]. The extended assessment leads us to some interesting findings and conclusions. We detail them in the section as follows.

### E.1.1 Automatic Metric Evaluation

We use SummaC to evaluate the generated summaries by the FactPEGASUS on the same 100 random samples as the ones used

---

| Error Type | CNNDM | | XSum | |
|---|---|---|---|---|
| | Fact PEGASUS | Dual AMs | Fact PEGASUS | Dual AMs |
| Entity intrinsic | 13 | 7 | 21 | 12 |
| Entity extrinsic | 0 | 7 | 0 | 1 |
| *Subtotal* | *13* | *14* | *21* | *13* |
| Modifier | 2 | 0 | 14 | 3 |
| Event | 0 | 2 | 5 | 9 |
| Event-time | 0 | 2 | 2 | 8 |
| Location | 0 | 1 | 7 | 10 |
| Negation | 0 | 2 | 1 | 0 |
| Number | 0 | 5 | 4 | 12 |
| Misspelling | 0 | 5 | 0 | 0 |
| Broken sentence | 44 | 0 | 2 | 0 |
| *Subtotal* | *46* | *17* | *35* | *42* |
| *Total* | *59* | *31* | *56* | *55* |

Table 11: Human evaluation of FactPEGASUS factuality on the 100 random samples (CNNDM and XSum).

in the main script. The FactPEGASUS results in $\mathbf{SummaC_{zs}}(\mu\text{-}85.0\%, \sigma\text{-}0.199)$ and $\mathbf{SummaC_{Conv}}(\mu\text{-}75.9\%, \sigma\text{-}0.193)$ on CNNDM while producing $\mathbf{SummaC_{zs}}(\mu\text{-}18.9\%, \sigma\text{-}0.309)$ and $\mathbf{SummaC_{Conv}}(\mu\text{-}27.4\%, \sigma\text{-}0.150)$ on XSum. The FactPEGASUS has better SummaC scores than ours while, on the other hand, our models present better ROUGE scores as listed in the main script. To better understand what underlies such a disconnected correlation between the ROUGE and SummaC scores, we also further conduct our human evaluation on the same 100 summaries as follows.

### E.1.2 Human Evaluation

During our evaluation, we notice that the FactPEGASUS-generated summaries from CNNDM contain many broken sentences. That is, the sentences are suddenly clipped in a way that they finish with words such as prepositions, pronouns, or determiners dangling at the end with missing information. We think that considering the phenomenon is important because the endings could otherwise avert various hallucination occurrences but can also miss out on important facts and leave the summaries in an incomplete and less comprehensible state. We thus include the assessment of the phenomenon. The results are shown in Table 11 where we compare the FactPEGASUS with the Dual AMs.

Compared to the FactPEGASUS on CNNDM, the Dual AMs method performs better on the intrinsic named entity-related hallucinations (NERHs)

---

[15] https://github.com/tingofurro/summac.

[16] It is worth noting that FactPEGASUS/Zero-shot uses FactCC to score important pseudo-summary sentence selection for masked pre-training and FactCC was trained with CNNDM-derived data. As we use the authors' published source code and dataset to fine-tune and evaluate FactPEGASUS zero-shot on CNNDM, the zero-shot requirement is thus relaxed.

while the FactPEGASUS has much fewer extrinsic entity hallucinations and syntactic agreement and misspelling errors. But the FactPEGASUS has a significant number of broken sentences. This could explain its lower ROUGE scores but higher SummaC's factuality consistency scores in that the clipping sentence phenomenon may degenerate the n-gram overlap matches. On the other hand, it could benefit the factual entailment assessment in latent semantic space because of less noise and fewer outliers, even though it might lose many facts at the same time. This can also partially explain its few extrinsic entity hallucinations.

Comparing the models on XSum, we see that the Dual AMs also performs much better on the reduction of intrinsic NERHs than the FactPEGASUS while the FactPEGASUS results in fewer syntactic agreement errors. Meanwhile, the FactPEGASUS has much higher errors in the modifier category. Given XSum has much shorter summaries discussed in the main script, the FactPEGASUS creates far fewer broken sentences than happens on CNNDM. Overall, the Dual AMs method shows outperformance on factuality consistency.

### E.1.3  Discussion

The analysis of the ROUGE results, SummaC scores, and human evaluation in main script has indicated that there is a correlational disparity between ROUGE scores and factuality consistency results. The assessment in Appendix E.1 provides us further insights into how various factual and syntactic errors might underpin the ROUGE and SummaC scores. We see how the generated summaries with the clipping phenomenon can result in low n-gram overlap-based metric scores but high factual entailment metric results. It indicates to us that the automatic metrics can be a 'bias' in the model's generative capacity by not knowing the underlying conditions and phenomena of the generated summaries. We believe that the ATS factuality research going forward needs a systemic approach to understand the correlations of various quantitative metric scores concerning a wide range of factual and syntactic phenomena. Gaining such a holistic understanding of the correlations is going to open us up to new approaches to factuality modeling and evaluation metric development.

| Model | SummaC$_{ZS}$ | | SummaC$_{Conv}$ | |
|---|---|---|---|---|
| | $\mu(\%)$ | $\sigma$ | $\mu(\%)$ | $\sigma$ |
| BART-base | 72.5 | 0.233 | 68.8 | 0.214 |
| Dual AMs | **73.1** | 0.231 | **69.3** | 0.210 |

Table 12: SummaC score distribution statistics over the full CNNDM test set (11488 samples).

| Model | SummaC$_{ZS}$ | | SummaC$_{Conv}$ | |
|---|---|---|---|---|
| | $\mu(\%)$ | $\sigma$ | $\mu(\%)$ | $\sigma$ |
| BART-base | 9.6 | 0.177 | 23.7 | 0.048 |
| Dual AMs | **9.8** | 0.178 | 23.7 | 0.048 |

Table 13: SummaC score distribution statistics over the full XSum test set (11328 samples).

| Model | CNNDM | |
|---|---|---|
| | SummaC$_{ZS}$ | SummaC$_{Conv}$ |
| BART-base vs BART-base/Dual AMs | 0.007 (<0.05) | 0.003 (<0.05) |

Table 14: Statistic significance (paired t-test) on SummaC scores resulted from the full test set (CNNDM).

| Model | XSum | |
|---|---|---|
| | SummaC$_{ZS}$ | SummaC$_{Conv}$ |
| BART-base vs BART-base/Dual AMs | 0.304 (>0.05) | 0.912 (>0.05) |

Table 15: Statistic significance (paired t-test) on SummaC scores resulted from the full test set (XSum).

### E.2  Factuality Evaluation Statistical Significance Assessment

As devised to reduce intrinsic entity hallucinations, our methods have different objectives from n-gram matching seen in pre-training language modeling. So, we expect that our methods improve scores slightly on n-gram overlapping-based metrics (e.g. ROUGEs) by matching the de-hallucinated entities and references in summaries. Given entities and references are sparse and small sets in long documents, we also expect that the improvement in overall automatic factuality scores would be small. But we would nonetheless like to evaluate SummaC on the full test sets and examine the statistical significance of SummaC scores between the BART-base backbone (trained with the Dual AMs) and the BART-base baseline. We first compute SummaC scores on the full test sets. Their score distributions are shown in Table 12 on CNNDM and Table 13 on XSum respectively. We then, using paired t-test[17], compute statistical significance on the SummaC scores between the BART-base backbone and the BART-base baseline, as shown in Table 14 on CNNDM and Table 15 on XSum respectively. Our null

---

[17]We use ttest_rel API from Python's scipy.stats package.

hypothesis is that there is no significant difference between scores on the generated summaries from the two models. We set the significance level at 0.05 as standard.

By rejecting the null hypothesis, Table 14 indicates that the scores on the generated summaries from the Dual AMs-trained backbone are significantly different from those from the baseline on CNNDM. Given Table 12 have shown that the backbone achieves better factuality scores than the baseline, this confirms our confidence that the summaries generated from the Dual AMs-trained backbone achieve significantly better SummaC scores than those from the baseline. On the other hand, Table 15 on XSum accepts the null hypothesis, and indicates that there is no significant difference in scores on the generated summaries between the Dual AMs-trained backbone and the baseline. We think that the results on XSum may be partly due to the one-sentence conciseness of XSum's summaries. Nonetheless, it agrees with the results in Table 13.

## F  Defining Rules for Entity-Related Hallucination Analysis

Named entity-related hallucinations (NERHs) can have various forms. Therefore, it is necessary to define rules to identify those hallucinations for our statistical analysis. Although the focus of this paper is the intrinsic NERHs, the identifying rules also cover the extrinsic NERHs. So, we categorize our assessment rules for a NERH as follows,

1. It is an intrinsic NERH if a named entity is mistaken for the other named entity within the context of the source document.

2. It is an intrinsic NERH if a named entity is mistaken concerning the context of the source document where there may not be any other named entity present in the source document.

3. A named entity has its name incorrect, for example, surname and given name. If the full name is wrong, we categorize it as an extrinsic NERH. If the partial name (either surname or given name) is wrong, we categorize it as an intrinsic NERH. However, if there are one or two wrong characters in a partial name, we categorize it as a misspelling instead.

4. It is an intrinsic NERH if a named entity's coreference is mistaken for another entity's

| Package | Version |
|---|---|
| Python | 3.8.10 |
| Pytorch | 1.11.0+cu113 |
| Hugging Face (Transformers) | 4.9.0 |
| Pytorch Geometric (PyG) | 2.0.4 |
| PyKeen | 1.9.0 |
| Rouge Metrics | 0.0.4 |
| Stanford CoreNLP | 4.4.0 |

Table 16: Key package versions.

coreference, for example, mistaking 'she' for 'he'.

5. It is an intrinsic NERH if a named entity is missing altogether when it is necessary in its place. For example, no named entity is present when a coreference (e.g., he or she) is mentioned.

We do not mark hallucinations for entity aggregations if their logical containment relation with the aggregated named entity is correct given their context.

As the pre-trained BART model has encoded a large amount of prior knowledge of general information, there are considerable amounts of extrinsic hallucinations that may be factual. Thus, we do not consider the cases as hallucinations or errors if we can verify them using external knowledge bases such as Wikipedia, online news, and/or Google Maps even though they cannot be deduced from the source document. We do not mark cases as errors if they can be reasonably deduced from the source documents. We extend the negation category to cover different forms of meaning contradiction at the phrasal level.

## G  Versions and Licenses

### G.1  Key Packages

#### G.1.1  Versions

We summarize the versions of key third-party packages used in our work in Table 16.

#### G.1.2  Licenses

The licenses related to these package versions are described as follows.

**Python**  Python license is accessible here[18].

---

[18] https://docs.python.org/3.8/license.html.

| Dataset | Version |
|---------|---------|
| CNNDM   | 3.0.0   |
| XSum    | 1.0.0   |

Table 17: Dataset versions.

**Pytorch**   Pytorch uses a collective license[19].

**Hugging Face**   Hugging Face[20] uses Apache-2.0 License, and covers its packages including Transformer derived models (e.g., BART), pre-trained BART-base tokenizer, datasets package (incl. CNNDM and XSum), and Accelerate package.

**PyG**   Pytorch Geometric[21] is a graph neural network modeling package and uses MIT License.

**PyKeen**   PyKeen[22] is a knowledge graph learning package and uses MIT License.

**Rouge Metrics**   Rouge Metrics uses Apache-2.0 License.

**Stanford CoreNLP**   CoreNLP[23] uses GNU General Public License v3.0.

**FactPEGASUS**   FactPEGASUS[24] uses MIT License.

**SummaC**   SummaC[25] uses Apache-2.0 License.

### G.2   Datasets

Table 17 lists version information of both CNNDM and XSum datasets used in our work.

---

[19]https://github.com/pytorch/pytorch/blob/master/LICENSE.
[20]https://github.com/huggingface.
[21]https://github.com/pyg-team/pytorch_geometric.
[22]https://github.com/pykeen/pykeen.
[23]https://github.com/stanfordnlp/CoreNLP/tree/v4.4.0.
[24]https://github.com/meetdavidwan/factpegasus.
[25]https://github.com/tingofurro/summac.