# OpenReview forum: "Mitigating Intrinsic Named Entity-Related Hallucinations of Abstractive Text Summarization"
_EMNLP/2023/Conference — EMNLP 2023 Findings_

### Official Review · Reviewer_n18X · 2023-08-04

**Soundness:** 3

**Excitement:**

3: Ambivalent: It has merits (e.g., it reports state-of-the-art results, the idea is nice), but there are key weaknesses (e.g., it describes incremental work), and it can significantly benefit from another round of revision. However, I won't object to accepting it if my co-reviewers champion it.

**Paper Topic And Main Contributions:**

To mitigate the implicit entity-related hallucinations, this paper proposes an adaptive margin ranking loss function and adopts an entity-sentence alignment method (E-Sent AM) to tackle entity-entity hallucinations and an entity-reference alignment method (E-Ref AM) to mitigate entity-reference hallucinations. Experiments on the CNNDM and XSum datasets show that it improves the baseline model on automatic evaluation scores.

**Reasons To Accept:**

The paper is generally well written and the proposed method is easy to follow and understand.

**Reasons To Reject:**

1. The improvement of the proposed method over the baseline in terms of automatic evaluation metrics is not obvious, and further validation of the effectiveness of the proposed method in terms of producing informative summaries is needed.
2. In the manual evaluation, the authors counted the entity hallucinations, syntactic agreement errors, and misspelling errors respectively, why not further classify the intrinsic entity hallucinations more finely into the two intrinsic hallucinations proposed to be solved in this paper: i.e. the entity-entity hallucinations and the entity-reference hallucinations, in order to further argue whether the two alignment methods proposed in this paper are effective or not in mitigating the entity hallucinations.
3. For the analyses in Tables 5 and 6, there are several observations on the CNNDM and XSum datasets that are not discussed in detail by the authors.

**Reproducibility:**

4: Could mostly reproduce the results, but there may be some variation because of sample variance or minor variations in their interpretation of the protocol or method.

**Reviewer Confidence:**

3: Pretty sure, but there's a chance I missed something. Although I have a good feel for this area in general, I did not carefully check the paper's details, e.g., the math, experimental design, or novelty.

---

> ### Author Rebuttal · Authors · 2023-08-28
>
> Thank you for your efforts to review our paper and for providing us with your expert feedback.
>
> *The improvement of the proposed method over the baseline in terms of automatic evaluation metrics is not obvious.*
> - As devised to reduce intrinsic entity hallucinations, our methods have different objectives from n-gram matching seen in pre-training language modeling. So, we expect that our methods improve scores slightly on n-gram overlapping-based metrics (e.g. ROUGEs) by matching the de-hallucinated entities and references in summaries.
>     Given entities and references are sparse and small sets in long documents, we also expect that the improvement in overall automatic factuality scores would be small. These factors motivated us to conduct the manual evaluation of the effectiveness of our objectives on entity-related hallucinations and other categories. The manual evaluation indicated that our objectives are complementarily effective on their goals.
>     Following prior works (e.g., [lyu_faithful_2022]; [kouris_abstractive_2021]; [zhu_enhancing_2021]) which may or may not improve automatic metric scores given the merits of their objectives, we would highlight our results with respect to our objectives.
>
> *Why not further classify the intrinsic entity hallucinations more finely into the two proposed methods?*
> - This paper did not separately evaluate entity-entity hallucinations and entity-reference hallucinations because the evaluation of 100 random samples did not see obvious entity-reference hallucinations. Additionally, the results from the manual evaluation showed that two alignment methods are complementary to reduce intrinsic entity-related hallucinations more than individual methods working in isolation.
>
> *Several observations on the CNNDM and XSum datasets in Tables 5 and 6 are not discussed in detail.*
> - We strived to reveal the most valuable aspects of our work, including observations and insights, within the 8-page limit. So, we may omit some discussions that may be less important. In this case, we focus on entity-related hallucinations, most error occurrences of other categories, and their plausible causes that we might be able to explain from Tables 5 and 6.
>
>
> [lyu_faithful_2022]: Lyu, Yuanjie and Zhu, Chen and Xu, Tong and Yin, Zikai and Chen, Enhong, "Faithful Abstractive Summarization via Fact-aware Consistency-constrained Transformer", {Proceedings of the 31st ACM International Conference on Information \& Knowledge Management}, pp.1410--1419, October 2022.
>
> [kouris_abstractive_2021]: Kouris, Panagiotis and Alexandridis, Georgios and Stafylopatis, Andreas, "Abstractive Text Summarization: Enhancing Sequence-to-Sequence Models Using Word Sense Disambiguation and Semantic Content Generalization", {Computational Linguistics}, vol.47, pp.813--859, December 2021.
>
> [zhu_enhancing_2021]: Zhu, Chenguang and Hinthorn, William and Xu, Ruochen and Zeng, Qingkai and Zeng, Michael and Huang, Xuedong and Jiang, Meng, "Enhancing Factual Consistency of Abstractive Summarization", {Proceedings of the 2021 Conference of the North American Chapter of the Association for Computational Linguistics: Human Language Technologies}, pp.718--733, June 2021.

---

### Official Review · Reviewer_oSii · 2023-08-14

**Typos Grammar Style And Presentation Improvements:** 1. Line 20,25 ~used
2. In line 256, d…
**Soundness:** 3

**Excitement:**

4: Strong: This paper deepens the understanding of some phenomenon or lowers the barriers to an existing research direction.

**Paper Topic And Main Contributions:**

The paper addresses the problem of hallucination in abstractive summarization. Hallucination refers to the spans in the generated output that either are not supported (extrinsic) or contradict(intrinsic) the input context. Particularly, the authors propose a loss function named adaptive margin ranking loss to reduce entity based intrinsic hallucination. They propose two loss functions, namely, entity-sentence alignment and entity-reference alignment methods. The Entity-sentence alignment method takes into account the entity names while the latter takes into account its reference pronouns in the remainder of the article. They use contrastive learning by sampling positive and negative samples from the entity clusters. They adapt the Distance-IoU from [1] and propose Entity span Distance-IoU for the loss function.

References:
1. Zhaohui Zheng, Ping Wang, Wei Liu, Jinze Li, Rong guang Ye, and Dongwei Ren. 2020. Distance- IoU Loss: Faster and Better Learning for Bounding Box Regression. Proceedings of the AAAI Conference on Artificial Intelligence, 34(07):12993 13000. Number: 07.


**Questions For The Authors:**

A. It is unclear how we learn the token representation in eq 7?

B. Please bold the highest scores in Table 1 and 2.

C. In Table 1 and 2, compared to the BART-base, can you please verify if the improvements attained in the proposed methods are statistically significant?

D. The evaluation needs to be done on the whole test set to accurately reflect the performance of the models. It would be better to re-compute the Table 3 and 4 to verify if the trend correlates for the whole test set.

E. In Table 4, $SummaC_{Conv}$ scores, the scores barely change in the third decimal point. 0.233 to 0.239. Please verify if its statistically significant.

Might increase the scores after re-computing them for Table 3 and 4. (answer to Question D).

**Reasons To Accept:**

1. The paper has novel contribution for entity based intrinsic hallucination area.
2. Human evaluation of the randomly sampled generated summaries provide more insights into the type of entity hallucination errors the baseline makes compared to the proposed method.
3. Compares the proposed model's performance based multiple baseline SOTA.

**Reasons To Reject:**

1. It is unclear why the authors disregard mutually excluding context-guided learning approach while performing negative sampler. Providing intuition for this might benefit the overall paper.
2. It might be easier to understand the methodology by providing explanation on how the entity clusters and their corresponding references are constructed.
3. It is unclear whether the results obtained in Table 1-4 are significant compared to the baseline.
4. The paper randomly samples 100 examples from the test set to evaluate their faithfulness. The relatively high standard deviation around 0.22 (where score range is 0 to 1), tells that sampling 100 instances is not enough to conclude of the performance. It might be better to re-compute the Table 3 and 4 with whole test set to verify whether the trend still stands.

**Reproducibility:**

4: Could mostly reproduce the results, but there may be some variation because of sample variance or minor variations in their interpretation of the protocol or method.

**Reviewer Confidence:**

3: Pretty sure, but there's a chance I missed something. Although I have a good feel for this area in general, I did not carefully check the paper's details, e.g., the math, experimental design, or novelty.

---

> ### Author Rebuttal · Authors · 2023-08-28
>
> Thank you for your efforts to review our paper and for providing us with your expert feedback.
>
> *Unclear why the authors disregard mutually excluding context-guided learning approach while performing a negative sampler.*
> - Thank you for raising the valid question. We design the context guidance approach instead of mutual exclusion sampling for the following considerations, a) We aim to maintain sufficient data samples to generalize trained models better. b) The context guidance approach also aims at helping models learn differences from similarities. It is analogous to human abstraction.
>
> *Provide explanation on how the entity clusters and their corresponding references are constructed.*
> - We use the Stanford CoreNLP parsing tool to produce conference resolution and sentence construction.
>     Conference resolution consists of named entities and their references. Each named entity is clustered with its references. Both entities and references are annotated with their positions concerning their sentences, the associated sentence number, and several relation attributes.
>     We also compute each sentence's length and the total number of sentences.
>     These are word-level annotations.
>     We also use the BART model's tokenizer (using BPE tokenization) to construct the token sequence of the document, and word-token map during the preprocessing.
>     The word-token map allows runtime mapping between word indices and token indices.
>     So, using data from the preprocessing, we build training triplet samples of entity, reference, and relation along with their positions from runtime mini-batches.
>     The related underlying implementations are mainly detailed in Appendix D.1, D.2.1, D.2.6, and C.
>     We are to improve on these appendices following your suggestion.
>
> *It is unclear how we learn the token representation in eq 7?*
> - With data from the point above, we can recover sentence token indices with respect to a document concerning equation 4 and equation 5. We then use the token indices to gather (i.e. index) the corresponding latent outputs of the encoder for equation 7.
>
> *The evaluation needs to be done on the whole test set.*
> - Following your advice, we updated SummaC evaluation on the full tests.
>     On CNNDM (11488 samples), BART-base baseline results in 72.5\%($\mu$) and 0.233($\sigma$) on SummaCzs, and 68.8\%($\mu$) and 0.214($\sigma$) on SummaCconv. Dual AMs results in 73.1\%($\mu$) and 0.231($\sigma$) on SummaCzs, and 69.3\%($\mu$) and 0.210($\sigma$) on SummaCconv.
>     On XSum (11328 samples), BART-base baseline results in 9.6\%($\mu$) and 0.177($\sigma$) on SummaCzs, and 23.7\%($\mu$) and 0.048($\sigma$) on SummaCconv. Dual AMs results in 9.8\%($\mu$) and 0.178($\sigma$) on SummaCzs, and 23.7\%($\mu$) and 0.048 ($\sigma$) on SummaCconv.
>     The results are consistent with those evaluated on 100 random samples in the paper.
>
> *Verify statistical significance of SummaCconv in Table 4 and Rouges in Tables 1 and 2.*
> - We computed statistical significance on SummaC scores using a paired t-test between BART-base baseline and Dual AMs following your suggestion. The null hypothesis is no significant difference between them.
>     On CNNDM, we got a p-value of 0.007 (<0.05) for SummaCzs, and 0.003 (<0.05) for SummaCconv.
>     On XSum, we got a p-value of 0.304 (>0.05) for SummaCzs, and 0.912 (>0.05) for SummaCconv.
>     Results for CNNDM reject the null hypothesis while the results for XSum accept it. This may be partly due to the one-sentence conciseness of XSum's summaries.
>     Following your advice, we further computed a paired t-test on ROUGE f-scores between BART-base baseline and Dual AMs.
>     On CNNDM, p-value is 0.9443 for rouge-1, 0.9820 for rouge-2, and 0.8172 for rouge-L.
>     On XSum, p-value is 0.7119 for rouge-1, 0.3968 for rouge-2, and 0.3281 for rouge-L.
>     As devised to reduce intrinsic entity hallucinations, our methods have different objectives from n-gram matching seen in pre-training language modeling. So, we expect that our methods improve scores slightly on n-gram overlapping-based metrics (e.g. ROUGEs) by matching the de-hallucinated entities and references in summaries.
>     Given entities and references are sparse and small sets in long documents, we also expect that the improvement in overall automatic factuality scores would be small. These factors motivated us to conduct the manual evaluation of the effectiveness of our objectives on entity-related hallucinations and other categories. The manual evaluation indicated that our objectives are complementarily effective on their goals.
>     Following prior works (e.g., [lyu_faithful_2022]; [kouris_abstractive_2021]; [zhu_enhancing_2021]) which may or may not improve automatic metric scores given the merits of their objectives, we would highlight our results with respect to our objectives.
>
> *Typos, and bold the highest scores in Table 1 and 2.*
> - Thank you for pointing out the typos. We are to fix them. We are also to bold the highest scores in Table 1 and Table 2 following your advice.
>
> [lyu_faithful_2022]: Lyu, Yuanjie and Zhu, Chen and Xu, Tong and Yin, Zikai and Chen, Enhong, "Faithful Abstractive Summarization via Fact-aware Consistency-constrained Transformer", {Proceedings of the 31st ACM International Conference on Information \& Knowledge Management}, pp.1410--1419, October 2022.
>
> [kouris_abstractive_2021]: Kouris, Panagiotis and Alexandridis, Georgios and Stafylopatis, Andreas, "Abstractive Text Summarization: Enhancing Sequence-to-Sequence Models Using Word Sense Disambiguation and Semantic Content Generalization", {Computational Linguistics}, vol.47, pp.813--859, December 2021.
>
> [zhu_enhancing_2021]: Zhu, Chenguang and Hinthorn, William and Xu, Ruochen and Zeng, Qingkai and Zeng, Michael and Huang, Xuedong and Jiang, Meng, "Enhancing Factual Consistency of Abstractive Summarization", {Proceedings of the 2021 Conference of the North American Chapter of the Association for Computational Linguistics: Human Language Technologies}, pp.718--733, June 2021.

---

### Official Review · Reviewer_sW6c · 2023-08-14

**Soundness:** 4

**Excitement:**

3: Ambivalent: It has merits (e.g., it reports state-of-the-art results, the idea is nice), but there are key weaknesses (e.g., it describes incremental work), and it can significantly benefit from another round of revision. However, I won't object to accepting it if my co-reviewers champion it.

**Paper Topic And Main Contributions:**

The paper studies named-entity related hallucinations (NERHs) in the context of abstractive text summarization. The authors propose two extensions to a traditional Seq2Seq model trained using XE loss: 1) adaptive margin ranking loss function which includes adaptive scaling variable of the base margin, 2) entity-sentence and entity-reference alignments methods which mitigate the entity-context and entity-reference hallucinations respectively. Experiments are conducted two datasets CNNDM and XSum, using a BART-base architecture.

**Questions For The Authors:**

How does this method generalize to other domains (dialogue, scientific documents, etc) and model types (BART-large, decoder-only models, small (7B) foundation models)?
Some of the terminology and explanations seem unnecessarily complex, could the writing be simplified?

**Reasons To Accept:**

- Work touches on a very relevant and important topic of improving factuality
- Analysis showing promising results
- Well documented experiment details (Appendix)

**Reasons To Reject:**

- Limited scope of exploration leaves the question of generalizability of the method open:
a) Datasets used in study cover the same domain (news), does the method work for other domains, such as dialogue?
b) Model used in study is small by todays standards, does the method bring improvements to other architectures, such as decoder-only?
- Given that the authors had access to state-of-the-art compute (GPUs with 80GB memory) the choice of baselines seems insufficient

**Reproducibility:**

4: Could mostly reproduce the results, but there may be some variation because of sample variance or minor variations in their interpretation of the protocol or method.

**Reviewer Confidence:**

4: Quite sure. I tried to check the important points carefully. It's unlikely, though conceivable, that I missed something that should affect my ratings.

---

> ### Author Rebuttal · Authors · 2023-08-28
>
> Thank you for your efforts to review our paper and for providing us with your expert feedback.
>
> *Limited scope of exploration.*
> - We follow the lead of recent prior works to focus solely on abstractive text summarization based on the encoder-decoder architecture with the two most used datasets (CNNDM and XSum) since abstractive text summarization is a relatively big topic.
>     Meanwhile, it is a very valid suggestion to investigate the generalizability of our methods for other domains and architectures in our future work, such as abstractive text summarization using decoder-only architectures and abstractive dialogue summarization.
>
> *With GPUs with 80GB memory, the choice of baselines seems insufficient.*
> - The used GPUs/80GB in this paper are shared resources. It is infeasible for us to fully utilize them for training over several days at any given time. Meanwhile, small-sized pre-trained models such as pre-trained BART-base are used and accepted for abstractive text summarization research in several recent prior works including the cited prior works in this paper.
>     As our objectives in this paper are to address specific hallucinations instead of competing overall performance, we think that the BART-base is a reasonable choice for the purpose.
>
> *Some of the terminology and explanations could be simplified.*
> - Thank you for suggesting an improvement to some of the terminology and explanations. We are to go through a second round of proofreading to identify and simplify terminology and explanations.

---

### Official Review · Reviewer_VoaE · 2023-08-17

**Soundness:** 2

**Excitement:**

3: Ambivalent: It has merits (e.g., it reports state-of-the-art results, the idea is nice), but there are key weaknesses (e.g., it describes incremental work), and it can significantly benefit from another round of revision. However, I won't object to accepting it if my co-reviewers champion it.

**Paper Topic And Main Contributions:**

This work introduces an adaptive margin ranking that facilitates two entity-alignment learning methods for reducing intrinsic named entity-related hallucinations on abstraction text summarization tasks. Compared with BART-base, the experiment results show their proposed method with BART-base works on automatic and human evaluation metrics.

**Reasons To Accept:**

* These authors propose entity-sentence alignment adaptive margin ranking loss to help the model find more precise sentence spans of a certain entity.
* The authors propose an entity-reference alignment adaptive margin ranking loss for better entity-reference alignment.
* The results show the proposed methods achieve higher scores on SummaCzs and SummaCconv.


**Reasons To Reject:**

BART model is the only baseline on SummaCzs and SummaCconv, which is over 4 years old at this point, the model used BART-base is a weaker one. In addition, the improvements are relatively small and based on only 100 samples. It's not clear if these gains would hold up if the experiments are conducted on the full test sets on CNN/DM and XSum.

In detail:
1. Unfair Comparison. I don't understand why the authors use zero-shot FactPEGASUS as one of the baselines in Table 2. ROUGE evaluation.
2. Need more factuality evaluation metrics. The results on SummaCx are confusing since references have lower scores but the optimization object is higher scores. Therefore, I suggest authors use more factuality evaluation metrics such as FactCC and DAE.
3. The experimental results may be influenced by random samples. Only 100 samples are evaluated, and the gain is sight. I highly recommend evaluating the full test sets with different factuality evaluation metrics.
4. Original FactPEGASUS has significantly higher SummaCx scores (in Appendix E.1) than the proposed methods. Why not conduct the methods on FactPEGASUS and use it as another strong baseline?

**Reproducibility:**

4: Could mostly reproduce the results, but there may be some variation because of sample variance or minor variations in their interpretation of the protocol or method.

**Reviewer Confidence:**

4: Quite sure. I tried to check the important points carefully. It's unlikely, though conceivable, that I missed something that should affect my ratings.

---

> ### Author Rebuttal · Authors · 2023-08-28
>
> Thank you for your efforts to review our paper and for providing us with your expert feedback.
>
> *Unfair comparison to zero-shot FactPEGASUS as one of the baselines.*
> - We included FactPEGASUS as a baseline because
>     a) We use the authors' published source code and dataset to train and evaluate FactPEGASUS zero-shot on CNNDM. As FactPEGASUS uses FactCC to score important pseudo-summary sentence selection for masked pre-training and FactCC was trained with CNNDM-derived data, the zero-shot requirement is relaxed. The subsequently observed broken sentence phenomenon from the evaluation has ramifications for factuality evaluation metric research. We think that including it could be beneficial since the phenomenon has not been discussed elsewhere to the best of our knowledge.
>     b) The authors also pointed out in footnote 5 that the zero-shot requirement was relaxed to learn the connector used in their zero-shot evaluation.
>
> *The results on SummaCx are confusing since references have lower scores than the optimization object.*
> - Thank you for pointing out the less intuitive SummaC score disparity between gold references and the generated summaries. We think that it is a phenomenon mainly attributed to the extrinsic information in gold references ([gehrmann_repairing_2023]). Such a phenomenon is also observed in other factuality metric evaluations. For example, factuality evaluation scores on XSum are commonly much lower than those on CNNDM as seen in several prior works (e.g., [lyu_faithful_2022]; [dong_multi-fact_2020]; [yuan_bartscore_2021]) because XSum’s gold references contain much more extrinsic information than CNNDM’s ([lu_multi-xscience_2020]).
>
> *Need more factuality evaluation metrics.*
> - We strived to use evaluation tools from more recent research which often improve on several prior works. SummaC has taken advantage of several benchmark datasets of prior factuality evaluation metrics including FactCC. It has shown better performance than several prior works including DAE and FactCC-CLS on the benchmark datasets. A second factor in choosing evaluation tools is their package availability or their published source code working.
>
> *BART model is over 4 years old at this point, and BART-base is a weaker one.*
> - BART-derived encoder-decoders remained a popular choice for abstractive text summarization research. Depending on computational resource availability, recent prior works including the cited baselines in this paper may utilize BART-base. The GPUs/80GB used in this paper are shared resources such that it is infeasible for us to fully utilize them to train our methods with a large model (e.g., BART-large) over several days at any given time.
>
> *Why not conduct the methods on FactPEGASUS and use it as another strong baseline?*
> - We backbone a pre-trained model that has not been fine-tuned with any other downstream task. This allows us to assess the results of our methods (good or bad) knowing that there is little interference by a different downstream task of a prior work. Meanwhile, it is a great idea to use FactPEGASUS as a backbone to train a stronger FactPEGASUS with our methods in our future work. We can also follow this idea for other prior works.
>
> *Highly recommend evaluating the full test sets with different factuality evaluation metrics.*
> - We update the SummaC evaluation on the full tests.
>     On CNNDM (11488 samples), BART-base baseline results in 72.5\%($\mu$) and 0.233($\sigma$) on SummaCzs, and 68.8\%($\mu$) and 0.214($\sigma$) on SummaCconv. Dual AMs results in 73.1\%($\mu$) and 0.231($\sigma$) on SummaCzs, and 69.3\%($\mu$) and 0.210($\sigma$) on SummaCconv.
>     On XSum (11328 samples), BART-base baseline results in 9.6\%($\mu$) and 0.177($\sigma$) on SummaCzs, and 23.7\%($\mu$) and 0.048($\sigma$) on SummaCconv. Dual AMs results in 9.8\%($\mu$) and 0.178($\sigma$) on SummaCzs, and 23.7\%($\mu$) and 0.048 ($\sigma$) on SummaCconv.
>     Following your advice, we also managed to use QA-based QuestEval to evaluate summaries in time.
>     On CNNDM (11488 samples), BART-base baseline results in 57.9\%($\mu$) and 0.080($\sigma$). Dual AMs results in 58.0\%($\mu$) and 0.081($\sigma$).
>     On XSum (11328 samples), BART-base baseline results in 44.8\%($\mu$) and 0.128($\sigma$). Dual AMs results in 44.8\%($\mu$) and 0.127($\sigma$).
>     Given entities and references are sparse and small sets in long documents, we expect that the improvement in overall automatic factuality scores would be small. It is a factor that motivated us to conduct the manual evaluation of the effectiveness of our objectives on entity-related hallucinations and other categories. The manual evaluation indicated that our objectives are complementarily effective on their goals. Following prior works (e.g., [lyu_faithful_2022]; [kouris_abstractive_2021]; [zhu_enhancing_2021]) which may or may not improve automatic metric scores given the merits of their objectives, we would highlight our results with respect to our objectives.
>
>
> [gehrmann_repairing_2023]: Gehrmann, Sebastian and Clark, Elizabeth and Sellam, Thibault, "Repairing the Cracked Foundation: A Survey of Obstacles in Evaluation Practices for Generated Text", {Journal of Artificial Intelligence Research}, vol.77, pp.103--166, May 2023.
>
> [lyu_faithful_2022]: Lyu, Yuanjie and Zhu, Chen and Xu, Tong and Yin, Zikai and Chen, Enhong, "Faithful Abstractive Summarization via Fact-aware Consistency-constrained Transformer", {Proceedings of the 31st ACM International Conference on Information \& Knowledge Management}, pp.1410--1419, October 2022.
>
> [dong_multi-fact_2020]: Dong, Yue and Wang, Shuohang and Gan, Zhe and Cheng, Yu and Cheung, Jackie Chi Kit and Liu, Jingjing, "Multi-Fact Correction in Abstractive Text Summarization", {Proceedings of the 2020 Conference on Empirical Methods in Natural Language Processing (EMNLP)}, pp.9320--9331, November 2020.
>
> [yuan_bartscore_2021]: Yuan, Weizhe and Neubig, Graham and Liu, Pengfei, "BARTScore: Evaluating Generated Text as Text Generation", {Advances in Neural Information Processing Systems}, vol.34, pp.7263--27277, 2021.
>
> [lu_multi-xscience_2020]: Lu, Yao and Dong, Yue and Charlin, Laurent, "Multi-XScience: A Large-scale Dataset for Extreme Multi-document Summarization of Scientific Articles", {Proceedings of the 2020 Conference on Empirical Methods in Natural Language Processing (EMNLP)}, pp.8068--8074, November 2020.
>
> [kouris_abstractive_2021]: Kouris, Panagiotis and Alexandridis, Georgios and Stafylopatis, Andreas, "Abstractive Text Summarization: Enhancing Sequence-to-Sequence Models Using Word Sense Disambiguation and Semantic Content Generalization", {Computational Linguistics}, vol.47, pp.813--859, December 2021.
>
> [zhu_enhancing_2021]: Zhu, Chenguang and Hinthorn, William and Xu, Ruochen and Zeng, Qingkai and Zeng, Michael and Huang, Xuedong and Jiang, Meng, "Enhancing Factual Consistency of Abstractive Summarization", {Proceedings of the 2021 Conference of the North American Chapter of the Association for Computational Linguistics: Human Language Technologies}, pp.718--733, June 2021.

---

### Meta-Review · Area_Chair_2Sqa · 2023-09-19

**Recommendation:** 3

**Metareview:**

This paper proposes to address named entity-related hallucinations through an adaptive margin ranking loss function and two alignment methods 1)  entity-sentence alignment and 2) entity-reference alignment to reduce halluciation in abstractive text summarization. The results seem to be promising based on BART-base model evaluated on CNNDM and XSum.

Pros:
- Work touches on a very relevant and important topic of improving factuality
- Introduces novel adaptive margin ranking loss function and alignment methods.
- Analysis showing promising results
- Well-documented experiment details (Appendix)

Cons:
- The choice of baselines and datasets raises concerns about the generalizability of the proposed methods to other domains and larger, modern architectures.
- The performance improvements seen are relatively small and based on a limited number of samples (100), leading to questions about the significance of the results.
- "Given that the authors had access to state-of-the-art compute (GPUs with 80GB memory) the choice of baselines seems insufficient"

In general, reviewers express enthusiasm regarding the proposed methods. However, they pinpoint two significant concerns: the perceived weakness in the evaluation process and the unjustified choice of using BART-base as the backbone for the system. In its current form, while the submission retains some element of excitement, it does not meet the threshold for acceptance at the main conference. Addressing these concerns, possibly by incorporating other baseline models could potentially elevate the work to the required standard. On the other hand, I think the work meets the standard to be accepted in the findings.

I also found the justification of BART-base is not fully convincing to me, even bigger models does not work out, models like T5-small could be alternatives compared to test on a single model:

"With GPUs with 80GB memory, the choice of baselines seems insufficient.

The used GPUs/80GB in this paper are shared resources. It is infeasible for us to fully utilize them for training over several days at any given time. Meanwhile, small-sized pre-trained models such as pre-trained BART-base are used and accepted for abstractive text summarization research in several recent prior works including the cited prior works in this paper. As our objectives in this paper are to address specific hallucinations instead of competing overall performance, we think that the BART-base is a reasonable choice for the purpose.
"

---

### Decision · Program_Chairs · 2023-10-07

**Decision:**

Accept-Findings

**Comment:**

This paper proposes to address named entity-related hallucinations through an adaptive margin ranking loss function and two alignment methods 1)  entity-sentence alignment and 2) entity-reference alignment to reduce halluciation in abstractive text summarization. The results seem to be promising based on BART-base model evaluated on CNNDM and XSum.

Pros:
- Work touches on a very relevant and important topic of improving factuality
- Introduces novel adaptive margin ranking loss function and alignment methods.
- Analysis showing promising results
- Well-documented experiment details (Appendix)

Cons:
- The choice of baselines and datasets raises concerns about the generalizability of the proposed methods to other domains and larger, modern architectures.
- The performance improvements seen are relatively small and based on a limited number of samples (100), leading to questions about the significance of the results.
- "Given that the authors had access to state-of-the-art compute (GPUs with 80GB memory) the choice of baselines seems insufficient"

In general, reviewers express enthusiasm regarding the proposed methods. However, they pinpoint two significant concerns: the perceived weakness in the evaluation process and the unjustified choice of using BART-base as the backbone for the system. In its current form, while the submission retains some element of excitement, it does not meet the threshold for acceptance at the main conference. Addressing these concerns, possibly by incorporating other baseline models could potentially elevate the work to the required standard. On the other hand, I think the work meets the standard to be accepted in the findings.

I also found the justification of BART-base is not fully convincing to me, even bigger models does not work out, models like T5-small could be alternatives compared to test on a single model:

"With GPUs with 80GB memory, the choice of baselines seems insufficient.

The used GPUs/80GB in this paper are shared resources. It is infeasible for us to fully utilize them for training over several days at any given time. Meanwhile, small-sized pre-trained models such as pre-trained BART-base are used and accepted for abstractive text summarization research in several recent prior works including the cited prior works in this paper. As our objectives in this paper are to address specific hallucinations instead of competing overall performance, we think that the BART-base is a reasonable choice for the purpose.
"